# Highly Effective Thermally Activated Delayed Fluorescence Emitters Based on Symmetry and Asymmetry Nicotinonitrile Derivatives

**DOI:** 10.3390/molecules27238274

**Published:** 2022-11-27

**Authors:** Min Gyeong Choi, Chan Hee Lee, Chihaya Adachi, Sae Youn Lee

**Affiliations:** 1Department of Energy and Materials Engineering, Dongguk University, Seoul 04620, Republic of Korea; 2Center for Organic Photonics and Electronics Research (OPERA), Kyushu University, 744 Motooka, Nishi-ku, Fukuoka 819-0395, Japan; 3Department of Chemistry and Biochemistry, Kyushu University, 744 Motooka, Nishi-ku, Fukuoka 819-0395, Japan

**Keywords:** organic light-emitting diodes, thermally activated delayed fluorescence, nicotinocarbonitrile, indolocarbazole, symmetry

## Abstract

In this study, we developed two thermally activated delayed fluorescence (TADF) emitters, **ICzCN** and **ICzCYP**, to apply to organic light-emitting diodes (OLEDs). These emitters involve indolocarbazole (ICz) donor units and nicotinonitrile acceptor units with a twisted donor-acceptor-donor (D-A-D) structure for small singlet (S_1_) and triplet (T_1_) state energy gap (Δ*E*_ST_) to enable efficient exciton transfer from the T_1_ to the S_1_ state. Depending on the position of the cyano-substituent, **ICzCN** has a symmetric structure by introducing donor units at the 3,5-position of isonicotinonitrile, and **ICzCYP** has an asymmetric structure by introducing donor units at the 2,6-position of nicotinonitrile. These emitters have different properties, such as the maximum luminance (*L*_max_) value. The *L*_max_ of **ICzCN** reached over 10000 cd m^−2^. The external quantum efficiency (*η*_ext_) was 14.8% for **ICzCN** and 14.9% for **ICzCYP**, and both achieved a low turn-on voltage (*V*_on_) of less than 3.4 eV.

## 1. Introduction

Organic light-emitting diodes (OLEDs) have been researched actively as next-generation displays because of their advantages, such as flexibility, brightness, and light weight, since the first study in 1987 [1]. In recent decades, numerous studies about organic fluorescence and phosphorescence emitters in the visible region have been performed to enhance the internal quantum efficiency (*η*_int_) [2,3,4,5,6,7]. However, despite the exceptional stability and reliability of fluorescence materials, a low exciton production efficiency (*η*_ST_) of 25% of fluorescence materials in electrical excitation results in a low maximum *η*_int_ [8]. Phosphorescence emitters can achieve a maximum *η*_int_ of nearly 100% with singlet (S_1_) and triplet (T_1_) exciton harvesting via intersystem crossing (ISC) using heavy atom effect from transition metals in phosphorescence emitters [9,10]. Nevertheless, the high density of T_1_ excitons from a long radiative decay time causes a strong T_1_ exciton annihilation process and a significant efficiency decrease under high current density [11].

Recently, thermally activated delayed fluorescence (TADF) materials have been researched as alternatives to fluorescence and phosphorescence materials in OLEDs because of their high *η*_ST_ through reverse ISC (RISC) from the T_1_ to the S_1_ state, resulting in a maximum *η*_int_ of 100% [12,13,14,15,16]. A small S_1_ and T_1_ state energy gap (Δ*E*_ST_) attained by minimizing the overlap between the highest occupied molecular orbital (HOMO) and the lowest unoccupied molecular orbital (LUMO) can lead to efficient RISC. Accordingly, various studies have concentrated on molecular designs using twisted molecular frameworks between the donor and acceptor units to obtain a small Δ*E*_ST_ with spatially separated and promoted intramolecular charge transfer (ICT). Several donor units, such as phenoxazine [17] and acridine [18], are usually applied to achieve a high torsion angle between donor and acceptor units. The carbazole donor unit is also widely used for blue TADF emitters. However, a simple carbazole donor unit is insufficient because of its poor electron-donating ability and steric hindrance. In contrast, carbazole derivatives such as indolocarbazole (ICz) are promising donor units for blue TADF emitters because of their higher electron-donating ability and significant steric hindrance [19].

In this study, we developed two new TADF emitters, 3,5-bis(4-(5-phenylindolo [3,2-*a*]carbazole-12(5*H*)yl)phenyl)isonicotinonitrile (**ICzCN**) and 2,6-bis(4-(5-phenylindolo-[3,2-*a*]carbazole-12(5*H*)yl)phenyl)nicotinonitrile (**ICzCYP**). ICz donor and nicotinonitrile acceptor units were introduced to form twisted donor-acceptor-donor (D-A-D) molecular structures. **ICzCN** has symmetric D-A-D molecular structures, while **ICzCYP** has an asymmetric one because of the position of the introduced cyano-substituent in the acceptor unit. In this paper, we researched differences in characteristics according to symmetry from the position of the cyano-substituent in the acceptor unit.

## 2. Results and Discussion

### 2.1. Synthesis and Characterization

The synthesis procedure for obtaining **ICzCN** and **ICzCYP** is outlined in Appendix A. The reaction was conducted in an anhydrous solvent under a nitrogen atmosphere using Suzuki Miyaura coupling reactions between **2** and 3,5-dichloroisonicotinonitrile (for **ICzCN**) or 2,6-dichloronicotinocarbonitrile (for **ICzCYP**) [20,21]. Before measurements and device fabrication, temperature-gradient sublimation was performed to obtain high-purity materials, and the chemical structure of the final products was confirmed through ^1^H nuclear magnetic resonance (NMR) spectroscopy and matrix-assisted laser desorption/ionization time-of-flight (MALDI-TOF) mass spectroscopy analysis (Appendix A). Furthermore, the thermal properties of the materials were observed using differential scanning calorimetry (DSC) to measure the glass transition temperature and thermogravimetric analysis (TGA) to measure the decomposition temperature. No glass transition temperature was detected in the DSC analysis of emitters, and the decomposition temperatures of **ICzCN** and **ICzCYP** were 521 and 540 °C (Appendix A).

### 2.2. Density Functional Theory (DFT) Calculations

The density functional theory (DFT) calculation was executed to predict the characteristics of **ICzCN** and **ICzCYP** emitters. We confirmed that both emitters fulfill the conditions of TADF, including Δ*E*_ST_ below 0.3 eV, and confirmed the frontier molecular orbital contributions (FTO) and energy levels by performing time-dependent DFT (TD-DFT) calculations at the B3LYP/6-31G(d) level (Appendix A). As depicted in Figure 1a, the dihedral angles between phenyl linkers and cyanopyridine (*θ*_A-π_) of **ICzCN** are 48° and 55°, and those of **ICzCYP** were 19° and 31°. Furthermore, the dihedral angles between phenyl linkers and ICz units (*θ*_D-π_) of **ICzCN** and **ICzCYP** were 50–52° and 48–50°. Because the degree of twist between the ICz units and phenyl linker was high, HOMO and LUMO were well separated from each other in both emitters, and the Δ*E*_ST_ values of **ICzCN** and **ICzCYP** were 0.01 and 0.03 eV. Therefore, we estimated an improvement in the up-conversion rate from the T_1_ to the S_1_ state [22].

We also identified dipole moments and the orientation of the optimized emitters. **ICzCN** and **ICzCYP** had a dipole moment of 3.84 and 2.43 debye, and we approximated a stronger charge transfer in **ICzCN** than in **ICzCYP** (Appendix A). Furthermore, two emitters with different cyano-substituent positions exhibited different HOMO distributions of donor units. Compared with **ICzCN**, where HOMO is distributed in both donors, **ICzCYP** exhibited only a HOMO distribution in the 3-position of the acceptor. We confirmed that these differences would produce different characteristics in the two emitters.

### 2.3. Photophysical Properties

The ultraviolet-visible (UV-Vis) absorption and photoluminescence (PL) spectra of **ICzCN** and **ICzCYP** were measured in toluene. The absorption peaks (*λ*_abs_) of the two emitters representing a broad, weak peak are 371 nm for **ICzCN** and 370 nm for **ICzCYP** because of the ICT of electrons between the donor moiety and the acceptor moiety. The maximum emission peaks (*λ*_PL_) in toluene are 489 and 475 nm, all generating blue emissions (Figure 2). Furthermore, 12 wt% **ICzCN** and **ICzCYP** doped films in 2,8-bis(diphenylphosphineoxide)dibenzofuran (PPF) were fabricated to validate the photophysical properties of materials (Appendix A). PPF has a high T_1_ energy of 3.1 eV, which prevents exciton quenching from the T_1_ state of the TADF emitter to the host matrix [23].

**ICzCN** and **ICzCYP** doped film emitted sky-blue emission, with peaks of 491 and 484 nm, which were redshifted by 2 and 9 nm relative to those measured in toluene. The reason for this phenomenon is the interaction between the polar PPF host and the emitter [24]. Through measurement of the difference between the S_1_ energy level obtained from the onset of the fluorescence spectra measured at 77 K and the T_1_ energy level obtained from the onset of the phosphorescence spectra measured at 77 K, a small Δ*E*_ST_ for efficient RISC process was estimated as 0.06 eV for **ICzCN** and 0.05 eV for **ICzCYP** (Appendix A). The PL quantum yield (*Φ*_PL_) levels of 12 wt% **ICzCN** and **ICzCYP** doped films were measured to verify the ability of **ICzCN** and **ICzCYP** as an emitter, with values of 76 and 58% (Table 1 and Appendix A).

The decay lifetime of the prompt and delayed components in transient PL decay curves of 12 wt% **ICzCN** and **ICzCYP** doped films at 300 K were determined by the fitted triexponential model. The TADF characteristics of both emitters in temperature-dependent PL spectra were evident at 300 K (Figure 3 and Appendix A). The prompt decay lifetime (*τ*_p_) was obtained from the nano-second scale data, and the delayed decay lifetime (*τ*_d_) was obtained from the micro-second scale data. *τ*_p_ was 56 ns for **ICzCN** and 42 ns for **ICzCYP**, and *τ*_d_ was 14 μs for **ICzCN** and 31 μs for **ICzCYP**. The delayed decay lifetime of **ICzCYP** is longer than that of **ICzCN**, depending on the different positions of the cyano-substituent.

Furthermore, both emitters have a larger *Φ*_d_ than *Φ*_p_ because *k*_ISC_ is much larger than *k*_r_^S^. **ICzCN** achieved a high *Φ*_RISC_ of 84% compared with **ICzCYP** because **ICzCN** achieved a competitive *k*_RISC_ ratio to *k*_nr,T_ given its small *k*_nr,T_ value of 4.4 × 10^4^ s^−1^ [25,26]. The HOMO energy level also differed because of the position of the carbonitrile group in the molecule. The ionization energy (*I*_p_) obtained by photoelectron yield spectroscopy of neat film was 5.80 eV for **ICzCN** and 5.86 eV for **ICzCYP**, which affected the contribution of the donor based on the cyano-substituent position in emitters (Appendix A).

Moreover, all rate constants values except *k*_nr,T_ were greater than those of **ICzCYP** because of the small *τ*_p_, *τ*_d_ of **ICzCN**. The *k*_RISC_ was larger in **ICzCN** because of the structure in which both donors are located at the ortho-position of the cyano-substituent and the higher *Φ*_d_ to *Φ*_p_ ratio of **ICzCN** than **ICzCYP** [27,28]. Detailed rate constants are presented in Table 2, and the equation-of-rate constants are displayed in Appendix A.

### 2.4. Electroluminescence (EL) Performance

For measuring the electroluminescence (EL) performance of **ICzCN** and **ICzCYP** as TADF emitters, a device composed of indium tin oxide (ITO)/ 1,4,5,7,8,11-hexaazatriphenylene-hexacarbonitrile (HATCN), 10 nm/ *N*,*N*′-di(1-naphthyl)-*N*,*N*′-diphenyl-(1,1′-biphenyl)-4,4′-diamine (α-NPD, 30 nm)/1,3-bis(*N*-carbazolyl)benzene (mCP, 5 nm)/12 wt% **ICzCN**:PPF or **ICzCYP**:PPF emitter (30 nm)/PPF (5 nm) /2,2′,2″-(1,3,5-benzinetriyl)-tris(1-phenyl-1-*H*-benzimidazole (TPBi, 40 nm)/Lithium fluoride (LiF, 0.8 nm)/Aluminum (Al, 80 nm) was fabricated. In this structure, we used HATCN as a hole-injection layer (HIL), and α-NPD and TPBi as the hole-transporting layer (HTL) and electron-transporting layer (ETL). Furthermore, mCP and PPF served as exciton-blocking layers to prohibit T_1_ exciton quenching from the emitting layer to the hole-electron transporting layer [29].

The energy-level diagram of the TADF-OLEDs device of **ICzCN** and **ICzCYP** is confirmed in Figure 4a. The emission wavelength of devices in the EL spectra were 507 nm for **ICzCN** emitting green light and 497 nm for **ICzCYP** emitting sky-blue light (Figure 4b). The current-density-voltage-luminance (*J-V-L*) characteristics and external quantum efficiency (EQE, *η*_ext_) versus luminance of devices for **ICzCN** and **ICzCYP** are indicated in Figure 4c,d. These devices achieved low turn-on voltage (*V*_on_) of less than 3.4 V, and the maximum luminance (*L*_max_) values are 13742 cd m^−2^ for **ICzCN** and 7627 cd m^−2^ for **ICzCYP**. Furthermore, the reduction of EQE values compared with maximum *η*_ext_ was observed at 100 and 1000 cd m^−2^, and roll-off efficiencies were 23 and 43% for the device with **ICzCN** emitter and 55 and 81% for the device with **ICzCYP**. **ICzCN** achieved a high *k*_RISC_ value of 1.8 × 10^5^ s^−1^, which enabled efficient T_1_ exciton transfer and suppressed roll-off of device. While **ICzCYP** suffers from severe high roll-off efficiency because of the strong T_1_ exciton annihilation process attributed to the longer T_1_ exciton lifetime compare with **ICzCN** [30]. EL performance data of the devices are presented in Table 3.

## 3. Conclusions

In this study, we developed **ICzCN** and **ICzCYP** based on pyridinecarbonitrile and ICz as TADF emitters with symmetry and asymmetry of the acceptor unit according to the position change of the cyano-substituent. Both emitters exhibited small Δ*E*_ST_ values of 0.06 and 0.05 eV from effective HOMO and LUMO separation. **ICzCN** achieved a high *Φ*_RISC_ of 84% from a small *k*_nr,T_ of 4.4 × 10^4^ s^−1^, large *k*_RISC_ of 2.8 × 10^5^ s^−1^, and *k*_ISC_ of 3.0 × 10^7^ s^−1^, enabling efficient T_1_ exciton harvesting. Furthermore, devices realized a maximum *η*_ext_ of 14.8% for **ICzCN** and 14.9% for **ICzCYP** in green and sky-blue light emission. Based on these results, we confirmed the electron-accepting substituent position dependency of photophysical properties in TADF emitters and expect these findings to serve as a reference for future development of new TADF emitters.

## Figures and Tables

**Figure 1 molecules-27-08274-f001:**
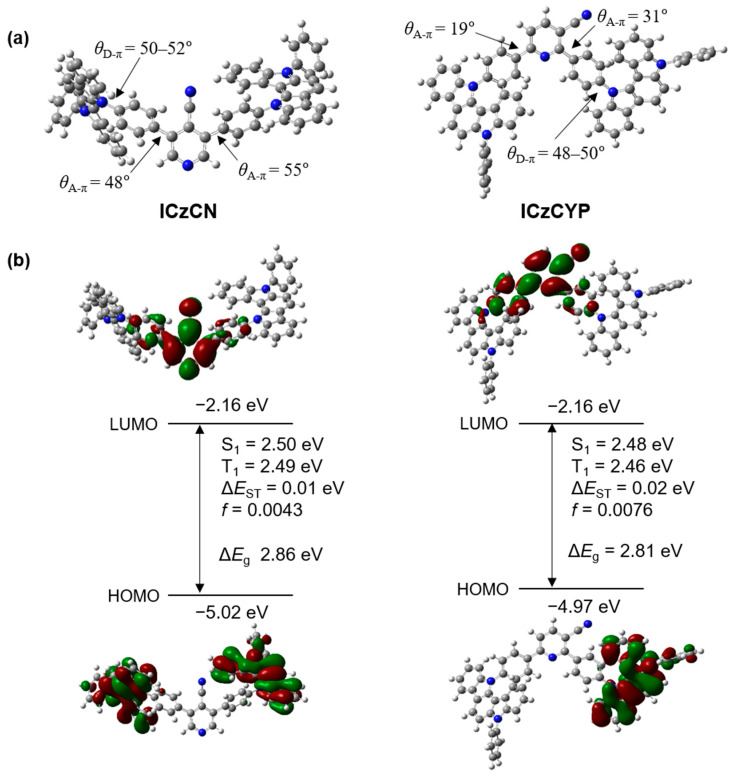
Optimized chemical structures and energy levels of S_1_ and T_1_ states and frontier orbital distributions of (**a**) **ICzCN** and (**b**) **ICzCYP**.

**Figure 2 molecules-27-08274-f002:**
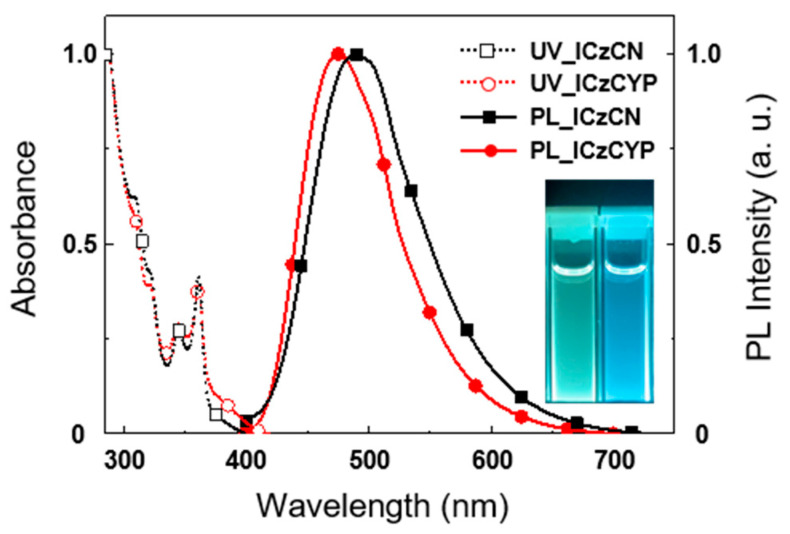
UV-Vis absorption and PL spectra of **ICzCN** and **ICzCYP** in toluene at room temperature. (inset: luminescence of **ICzCN** (left) and **ICzCYP** (right) in toluene.).

**Figure 3 molecules-27-08274-f003:**
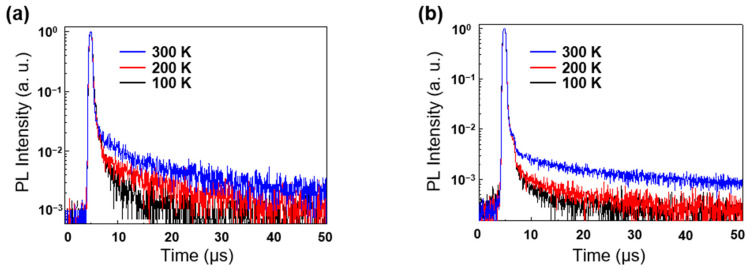
Temperature-dependent transient PL decay curves of (**a**) 12 wt% **ICzCN**:PPF and (**b**) **ICzCYP**:PPF doped films.

**Figure 4 molecules-27-08274-f004:**
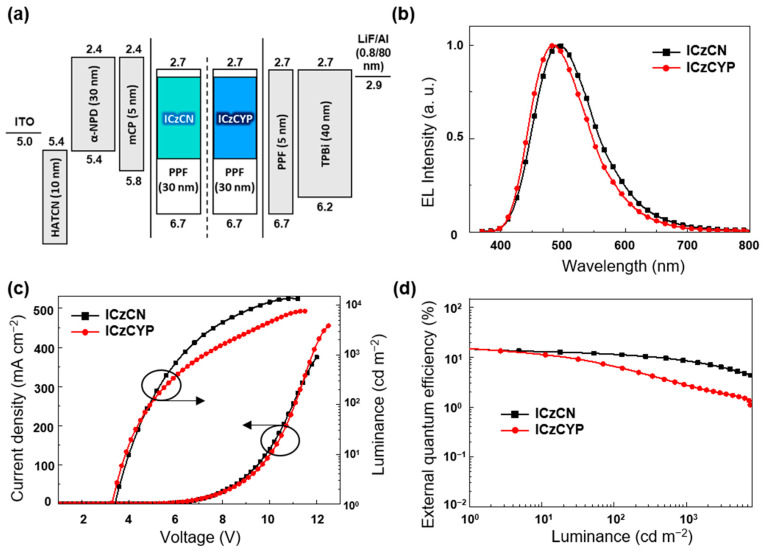
(**a**) Energy level diagram of a TADF-OLEDs device with **ICzCN** and **ICzCYP** as an emitter. (**b**) EL spectra of 12 wt% **ICzCN** and **ICzCYP** devices at 10 mA cm^−2^ (**c**) Current density-voltage-luminance (*J*-*V*-*L*) curves of TADF-OLEDs device with 12 wt% **ICzCN** and **ICzCYP** applied as emitter (**d**) External quantum efficiency versus luminance (*η*_ext_) of devices.

**Table 1 molecules-27-08274-t001:** Photophysical properties of **ICzCN** and **ICzCYP** emitters.

Emitter	*λ*_abs_^1^ (nm)	*λ*_PL_ [nm] sol ^1^/Film ^2^	*Φ*_PL_ [%] sol ^1^/Film ^2^	*τ*_p_^3^/*Φ*_p_^4^ (ns/%)	*τ*_d_^3^/*Φ*_d_^4^ (μs/%)	*I*_p_/*E*_a_/*E*_g_^5^ (eV)	*E*_S_/*E*_T_/Δ*E*_ST_^6^ (eV)
**ICzCN**	371	489/491	63/76	56/30	14/46	5.80/2.93/2.87	2.85/2.79/0.06
**ICzCYP**	370	475/484	52/58	42/27	31/ 31	5.86/2.95/2.91	2.88/2.83/0.05

^1^ Measured in dilute toluene solution (10^−5^ M) at room temperature. ^2^ 12 wt% doped film in PPF matrix. ^3^ Transient PL decay lifetime of prompt (*τ*_p_) and delayed (*τ*_d_) fluorescence for the 12 wt% doped films measured at 300 K. ^4^ Fractional contribution of prompt (*Φ*_p_) and delayed (*Φ*_d_) of the 12 wt% doped film measured at 300 K. ^5^ *I*_p_ was estimated from photoelectron yield spectroscopy, *E*_a_ is electron affinity and *E*_g_ was obtained by the onset of the absorption spectra of neat film. ^6^ S_1_ and T_1_ energies were obtained from the onset wavelengths in the PL spectra of 12 wt% doped films at 77 K.

**Table 2 molecules-27-08274-t002:** Photophysical properties of 12 wt% **ICzCN**:PPF and **ICzCYP**:PPF doped film.

Emitter	*k*_r_^S^ (s^−1^)	*k*_d_ (s^−1^)	*k*_nr, T_ (s^−1^)	*k*_ISC_ (s^−1^)	*k*_RISC_ (s^−1^)	*Φ*_ISC_ (%)	*Φ*_RISC_ (%)
**ICzCN**	1.3 × 10^7^	1.3 × 10^5^	4.4 × 10^4^	3.0 × 10^7^	2.8 × 10^5^	54	84
**ICzCYP**	6.7 × 10^6^	1.1 × 10^5^	6.4 × 10^4^	1.8 × 10^7^	1.8 × 10^5^	69	45

**Table 3 molecules-27-08274-t003:** EL performance data of 12 wt% **ICzCN** and **ICzCYP** devices.

Emitter	λ_EL_ ^1^ (nm)	*V*_on_^2^ (V)	*L*_max_ (cd m^−2^)	*η*_c_^3^ (cd A^−1^)	*η*_p_^3^ (lm W^−1^)	*η*_ext_^3^ (%)	CIE_x,y_ ^4^ (x,y)
**ICzCN**	507	3.4	13742	42.1	44.1	14.8	(0.26, 0.47)
**ICzCYP**	497	3.3	7627	46.5	50.3	14.9	(0.23, 0.45)

^1^ Maximum emission peak in EL spectra. ^2^ Turn-on voltage of devices at 1 cd m^−2^. ^3^ Maximum current efficiency (*η*_c_), maximum power efficiency (*η*_p_), and external quantum efficiency (*η*_ext_) ^4^ Commission Internationale de l’Éclairage (CIE) color coordination.

## Data Availability

Not applicable.

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
