# Peer review of "Highly Effective Thermally Activated Delayed Fluorescence Emitters Based on Symmetry and Asymmetry Nicotinonitrile Derivatives"

_molecules, 2022, doi:10.3390/molecules27238274_

Round 1

Reviewer 1 Report

In the manuscript, M. G. Choi et al. reported two thermally activated delayed fluorescence (TADF) emitters to apply to organic light-emitting diodes (OLEDs). The reported two molecules have a twisted donor-acceptor-donor structure for small single (S1) and triplet (T1) state energy gap (â–³EST) to enable efficient exciton transfer from the T1 to the S1 state. With different cyano-subsituent position, ICzCN and ICzCYP have the symmetric and asymmetric structures through introducing donor units at the corresponding positions. The results are interesting to some excent. I would like to recommend its publication in molecules if the comments mentioned below have been well addressed.

1.      Can author give more detailed understanding for the substituent-dependent OLED performance based on TADF, and provide a general guidance for further TADF emitter development?

2.      Finally, as a minor point, the English needs to be further improved.

Author Response

Point 1: Can author give more detailed understanding for the substituent-dependent OLED performance based on TADF, and provide a general guidance for further TADF emitter development?

Reponse 1: The reviewer asked to give more detailed understanding for the substituent-dependent OLED performance based on TADF, and provide a general guidance for further TADF emitter development. In this study, we confirmed symmetry dependency of TADF properties in newly designed donor-acceptor-donor (D-A-D) structure emitters. Symmetry structure ICzCN exhibited high PLQY, large kRISC compared with asymmetry structure ICzCYP and two times higher Lmax was observed in ICzCN even though EQEs of devices introducing two emitters were nearly same. These photo- or electro-physical properties difference between two emitters were originated from orbital configurations change with structural characteristics (Figure 1 (b)). Delocalized HOMO distribution in two indolocarbazole donor moieties of ICzCN enhance charge transfer and realize efficient TADF properties. In ICzCYP, localized HOMO in single donor moiety disrupt to effective charge transfer between donor and acceptor moieties and reduce intersystem crossing rate constant and PLQY more than ICzCN. Based on this result, we verified molecular design of symmetry structure TADF emitter with widely distributed HOMO is promising molecular design strategy to achieve effective photo- or electro-physical performances in OLEDs. And our previous research of structural characteristics and orbital configurations of TADF emitter was published in Nature Materials in 2015 (vol. 14, p330–336).

Point 2: Finally, as a minor point, the English needs to be further improved.

Response 2: Based on reviewers’ comment, we have carefully checked the grammar errors of our manuscript again.

Reviewer 2 Report

This report disclosed a novel study on two thermally activated delayed fluorescence (TADF) emitters, ICzCN and ICzCYP, to apply to organic light-emitting diodes (OLEDs). These emitters involve indolocarbazole (ICz) donor units and nicotinonitrile acceptor units with a twisted donor-acceptor- donor (D-A-D). The resultant device performance is somewhat good and the corresponding results and discussions are systematic and reasonable. For these reasons, it is suggested to be published in this journal. Some minor revisions are required.

1)     For Table 1, the delta EST, i.e. exchange energy, of ICzCN and ICzCYP, is experimentally obtained by measuring fluorescent spectra in 300K and phosphorescent spectra in 77K. Despite this method is widely used in references, it is not strictly right. I suggest the author measure them both at 77K.

2)     Please recheck the reference part. e.g. Ref. 18; ref. 22; ref. 25; ref. 26; ref. 30; ref. 37.

Author Response

Point 1:  For Table 1, the delta EST, i.e. exchange energy, of ICzCN and ICzCYP, is experimentally obtained by measuring fluorescent spectra in 300K and phosphorescent spectra in 77K. Despite this method is widely used in references, it is not strictly right. I suggest the author measure them both at 77K.

Response 1: Based on reviewers’ suggestion, we have fabricated co-deposited films of emitters and proceeded re-examination of fluorescent and phosphorescent spectrum in 77 K again. And all data of fluorescent and phosphorescent spectrum of manuscript, figures (Fig.S5 (c),(d)), and table (Table 1) were replaced and added by newly examined data.

Point 2: Please recheck the reference part. e.g. Ref. 18; ref. 22; ref. 25; ref. 26; ref. 30; ref. 37.

Response 2: Based on reviewers’ comment, we have carefully checked all references again and remove not suitable reference of number 17, 18, 22, 25, 26, 27, 28, 30, 36 and 37. And more proper references newly added in our manuscript (Revised reference number 22).